# New Symptoms in *Castanea sativa* Stands in Italy: Chestnut Mosaic Virus and Nutrient Deficiency

Sergio Murolo [1], Daniela Bertoldi [2], Federico Pedrazzoli [2], Manuela Mancini [1], Gianfranco Romanazzi [1] and Giorgio Maresi [2],*

1   Department of Agriculture, Food and Environmental Science, Marche Polytechnic University, Via Brecce Bianche, 60131 Ancona, Italy
2   Fondazione Edmund Mach, Centre for Technology Transfer, Via E. Mach 1, San Michele all'Adige, 38098 Trento, Italy
*   Correspondence: giorgio.maresi@fmach.it; Tel.: +39-046-1615-365

**Abstract:** The European chestnut characterizes both the landscape and economy of mountainous Italian areas. In recent years, new canopy disorders have been reported: "chestnut yellows", often ascribed to phytoplasma and/or nutrient deficiency, and "chestnut mosaic", associated with a virus (ChMV). Therefore, research was carried out in four Italian regions to describe the two symptomatic frames and assess their etiology. Surveys were conducted on 101 chestnut trees (23 with mosaic, 38 with yellowing, and 40 without symptoms). The phytosanitary status was monitored, and the new canopy disorders were detected, distinguishing between yellowing and mosaic. Moreover, leaf samples were collected for molecular and nutrient analyses. No phytoplasma infection was recorded, while ChMV was detected in 91.3% of samples with mosaic symptoms, 31.6% of yellowing samples, and 30.0% of asymptomatic samples. Yellowing was associated with Mn deficiency. On the other hand, ChMV-infected and healthy leaves had similar mineral contents, showing that mosaic symptoms are induced by the virus. Both disorders negatively affected photosynthesis efficiency. These phytosanitary problems are present in Italian chestnut woods and cause local effects, and a relationship with other biotic and abiotic factors can be hypothesized. Considering the increase in new records, these symptoms represent an emerging issue whose impact and spread need to be further monitored.

**Keywords:** crown yellowing; chestnut mosaic virus; SPAD measurements; chestnut wood; leaf nutrient content

## 1. Introduction

*Castanea sativa* (Mill.), the European chestnut, is the only European species of the genus *Castanea* and is largely widespread across the Mediterranean region, from the Black Sea to the Atlantic coast of the Iberian Peninsula [1]. In Italy, it covers approximately 780,000 hectares, mainly concentrated in the Apennines and Alps but also with a significant presence in the islands (Sardinia, Sicily, and Elba). This vast distribution of chestnut woods is the legacy of a long history of more than one thousand years during which chestnut orchards played a fundamental role in the survival of Italian mountain people, characterizing what is recognized as a "chestnut civilization" [2]: a way of life (or survival) strictly related to chestnut orchard cultivation and products. Currently, these widespread chestnut stands and orchards are still a fundamental asset for Italian mountains because of their multipurpose role, ranging from the production of timber and fruit to soil defense, water conservation, and forest heritage; furthermore, they allow the use of the territory for tourism and recreation and permit the survival of related biodiversity [3,4]. Chestnut continues to represent an essential economical source for areas where fruit production, mainly "marron" but also flour varieties, is still able to produce interesting incomes that

justify continuing their cultivation. In this context, chestnut wood also shows new and interesting perspectives.

Chestnut stands and especially orchards, strictly related to human maintenance, were able to survive the large depopulation that affected most of the Italian mountains after the 2nd World War [4,5]. Moreover, probably as a unique case for a tree species, they faced three biological invasions: ink disease starting in the 19th century [6,7]; chestnut blight in the middle of the 20th century [8], and Asian chestnut gall wasp at the beginning of the 21st century [9]. Biological control for the wasp, hypovirulence for blight, and the general resilience of the chestnut ecosystem against ink disease proved to be effective in the survival of the chestnut wood and the recovery or maintenance of the cultivation [5,10,11]. These phytosanitary factors played and still play a key role in chestnut management, severely affecting not only plants but also growers' and foresters' attitudes towards these woods. Therefore, any new problem causes concerns and doubts about the perseverance of cultivation, as the recent appearance (or reappearance) of fruit brown rot has confirmed [12,13]. In this context, despite an increasing interest in chestnut cultivation for fruit and wood production, new worries started for two symptomatologies: "chestnut yellows", described in different areas and often ascribed to phytoplasma [14–17] but also to a nutrient deficiency [18], and "chestnut mosaic", which was determined to be transmissible by grafting [14] and was recently associated with a virus [19–21] whose detection now is possible with molecular tools compared to a diagnosis based only on symptomatology in the past. Viral diseases have been underestimated for long period in the context of forests, but they can represent emergent phytosanitary problems in the near future, especially for forest tree species that are converted in specialized fruit orchards (i.e., chestnut). Unfortunately, detailed estimations of the economic losses induced by virus-diseased forest trees are lacking, but they could be comparable to those caused by viral infections on fruit trees [22].

Because of the concern regarding this problem, we studied and described the symptomatology, the possible causes, and the impact of the disease. This research aimed to (i) describe and discriminate the two symptomatic frames, defined as "mosaic" and "yellowing"; (ii) monitor the spread and impact of the two symptomatic frames in chestnut stands located in northern and central Italy; and (iii) verify the etiology of the two symptoms according to specific molecular tools and chemical analyses of leaves and soil.

## 2. Materials and Methods

### 2.1. Study Site

To localize affected stands, preliminary and informal interviews with leaders of chestnut associations, owners/managers, and local experts were conducted, showing pictures and asking directly if yellowing or foliage anomalies were observed. New contacts were added via snowball sampling [3,23,24]. In this way, sites along the Apennines and Pre-Alps in four different regions (Marche, Emilia-Romagna, Toscana, and Trentino-Alto Adige) were reported to have symptomatic trees and were chosen for the survey. For each site, indicated according to the initial letter of the region followed by a progressive number, the main features (location, province/region, geographical coordinates, elevation, aspect, substrate, stand type, symptomatic trees, and average tree age) were recorded during the surveys and are reported in Table 1. Most of the signaled sites were chestnut orchards, mainly cultivated with maroon, but symptomatic trees were also reported or observed during the survey in coppices or mixed wood; two nurseries reported yellowing in spring of 2021 and were thus included in the investigation.

### 2.2. Field Investigations

2.2.1. Phytosanitary Visual Assessment in the Plots

A visual survey was conducted on the selected areas in July 2021 to evaluate the symptoms associated with nutritional deficiency or chestnut mosaic disease and the presence of other phytosanitary problems related to Asian chestnut gall wasp, ink disease, and chestnut blight.

**Table 1.** The main features of the surveyed pre-Alpine and Apennine areas.

| Site ID | Location | Province/Region | Geographical Coordinates | Elevation m a.s.l. | Aspect | Substrate | Stand Type | Symptomatic Trees | Average Tree Age (y) |
|---|---|---|---|---|---|---|---|---|---|
| M1 | Venamartello | AP–Marche | 42,7882883; 13,4108881 | 657 | southeast | calcareous | orchard | grafted, sprouts | 40–50 |
| M2 | Acquasanta Terme | AP–Marche | 42.729.622; 13.403.105 | 918 | north west | calcareous | orchard | grafted, sprouts | 20–100 |
| ER1 | Selva la Maddalena | BO–Emilia Romagna | 44.240.029; 11.496.345 | 393 | north | marly siliceous | orchard | grafted | 100–200 |
| ER2 | Loiano | BO–Emilia Romagna | 44.253.954; 11.333.899 | 789 | west | siliceous | orchard | grafted | 20–50 |
| ER3 | Ca' di Balloni | BO–Emilia Romagna | 44.265.234; 11.339.537 | 653 | east | siliceous | orchard | grafted | 100 |
| ER4 | Gragnano | BO–Emilia Romagna | 44.245.631; 11.345.321 | 719 | northwest | siliceous | orchard | grafted, sprouts | 20 |
| ER5 | Malalbergo 1 | MO–Emilia Romagna | 44.223.022; 10.929.023 | 923 | northwest | siliceous | orchard | grafted | 10–200 |
| ER6 | Malalbergo 2 | MO–Emilia Romagna | 44.223.022; 10.929.023 | 923 | northwest | siliceous | orchard | grafted | 200 |
| ER7 | Alberelli | MO–Emilia Romagna | 44.231.896; 10.926.836 | 783 | west | siliceous | orchard | grafted | 100–200 |
| ER8 | Monteombraro Lavezze | MO–Emilia Romagna | 44.384.805; 10.994.774 | 628 | west | marly siliceous | orchard | grafted, sprouts | 20–30 |
| ER9 | Monteombraro Caseificio | MO–Emilia Romagna | 44.382.960; 11.007.369 | 664 | north | siliceous | new plantation | grafted | 5 |
| ER10 | Montese | MO–Emilia Romagna | 44.2471.51; 10.930.374 | 740 | west | siliceous | mixed wood | sprout | |
| TO1 | Castagneto | FI–Toscana | 43.937.377; 11.615.097 | 520 | southwest | siliceous | orchard, mixed wood | grafted, sprouts | 20–150 |
| TO2 | Gaggiolo | GR–Toscana | 42.889.658; 11.562.061 | 823 | northwest | siliceous | orchard | sprout (nonsymptomatic) | 20–150 |
| TN1 | Mezzolombarbo | TN–Trentino Alto Adige | 46.226.333; 11.072.695 | 328 | east | calcareous | orchard | grafted (nonsymptomatic) | 100 |
| TN2 | Drena | TN–Trentino Alto Adige | 45.962.994; 10.949.407 | 520 | west | calcareous | orchard | grafted | 100 |
| TN3 | Pranzo 1 | TN–Trentino Alto Adige | 45.919.926; 10.821.334 | 457 | east | calcareous | orchard, mixed wood | grafted, sprouts | 10–100 |
| TN4 | Pranzo 2 | TN–Trentino Alto Adige | 45.925.029; 10.815.196 | 537 | east | calcareous | orchard | grafted, sprouts | 10–100 |
| TN5 | Pranzo3 | TN–Trentino Alto Adige | 45.927.132; 10.814.403 | 590 | west | calcareous | orchard, mixed wood | grafted, sprouts | 10–100 |
| TN6 | Campi | TN–Trentino Alto Adige | 45.911.697; 10.817.839 | 670 | northwest | calcareous | orchard, mixed wood | grafted, sprouts | 10–100 |
| TN7 | Valterigo | TN–Trentino Alto Adige | 46.160.358; 11.163.589 | 711 | northwest | marly siliceous | new plantation | grafted | 5 |
| TN8 | Nursery San Michele | TN–Trentino Alto Adige | 46.203.289; 11.139.945 | 204 | west | calcareous | nursery | seedling | 1 |
| AD1 | Nursery Piccolongo | BZ–Trentino Alto Adige | 46.358.521; 11.284.808 | 231 | east | calcareous | nursery | seedling | 1 |

For each site, we defined a circle plot starting from a symptomatic plant (radius about 20 m; area = about 1300 m$^2$). Then, all the chestnut trees inside the area were monitored. The presence and positions of trees with anomalies in the color of the leaves (represented by mosaic, ascribable to viral infections, and/or yellowing symptoms) as well as leaf malformations (determined by nutrient deficiency) were recorded. The trees showing these symptoms were annotated on a two-dimensional map, and the total area and detailed pictures were taken to record the symptoms. For each site and inside the surveyed plot, we evaluated the intensity of phytosanitary problems according to an empirical scale ranging from 0 to 4: 0 = not present, 1 = sporadic presence (1–5 trees affected), 2 = medium presence (6–10 trees affected), 3 = high (11–15 trees), and 4 = very high (over 16 trees). The position of each affected tree was recorded using the Global Positioning System (GPS; e-Trex 30, Garmin, Olathe, KS, USA). The data obtained for individual trees were managed through Quantum Geographic Information System (GIS) open-source software and were displayed using Google Earth software.

A general survey of the area around the affected trees was also conducted considering a buffer zone of approximately 50 meters. As proposed by Pezzi et al., the presence of the three main diseases of chestnut was also evaluated on the selected trees [3]. Chestnut blight was assessed considering the relative abundance of healing, healed, virulent, and

intermediate cankers [10] on the crown and trunk. The predominance of hypovirulent (healing and healed cankers) or virulent infections was reported for each tree. Ink disease presence was checked by looking for classic symptoms: early symptoms (rarefied foliage and small and yellowing leaves) and dead trees with completely dead crowns and brown flames from the collar. These old and recent attacks were then recognized. Symptoms of Asian chestnut gall wasp (ACGW) attacks were also observed on the crown. Four classes were adopted for the parasite presence: no galls (absence), from 1 to 10 galls (sporadic), from 11 to 100 galls (presence), and more than 100 galls (heavy infestation). When possible, up to five galls were collected and opened to assess the presence of *Torymus sinensis*, the specific parasitoid of ACGW. The presence of other phytosanitary problems was also recorded considering foliage disease, such as powdery mildew and *Mycosphaerella maculiformis*, or other kinds of damage.

### 2.2.2. SPAD Measurements

In late August 2021, 28 representative trees showing symptoms of yellowing and mosaic and 19 asymptomatic trees were chosen in the study areas. In addition, four plants showing a partial recovery of symptoms compared to previous surveys were also considered. Plants were selected for their accessibility when trying to sample any side of the crown. Asymptomatic plants were chosen as near as possible to the symptomatic ones. The chlorophyll content was evaluated in each tree by an SPAD-502 chlorophyll meter (Konica Minolta sensing Inc., Sakai, Japan), a simple, portable diagnostic tool that measures the leaf transmittance in two wave bands (400–500 nm and 600–700 nm). The chlorophyll index provided by this device is proportional to the leaf chlorophyll concentration [25–28]. For each of the considered plants, 25 measures of SPAD were recorded (5 measures for each leaf and 5 leaves/plant).

### 2.2.3. Sample Collection

According to the bidimensional maps, we selected three different types of trees characterized by no symptoms (asymptomatic) (n. 40), mosaic (n. 23), and yellowing (n. 38). For each category, we collected leaf samples composed of ten shoots with at least ten leaves/shoots. All samples were put into separate plastic bags, which were stored in a refrigerator bag during transportation to the laboratory. The samples were subjected to a molecular analysis for the detection of phytoplasma and *Chestnut mosaic virus* (ChMV) and a chemical analysis for macro- and micronutrient determination, as described below. Twenty-three soil samples were collected during the surveys. Each sample consisted of a mixture of four subsamples collected all around the crown of the chosen affected trees. We collected approximately 1 kg of soil after removing the litter.

To record data on the macro- and micronutrient contents in chestnut leaves during the vegetative season, leaves were collected monthly from June to October 2021 for a total of five sampling points from three healthy trees located in three different areas of Trentino located in Mezzolombardo, Faedo, and Drena.

### 2.3. Laboratory Tests

#### 2.3.1. Molecular Detection of Phytoplasma and Chestnut Mosaic Virus in Chestnut Samples

For the detection of ChMV, the whole leaf blade was used, while phloem tissue isolated from each leaf by cutting midribs with a sterile scalpel was used as a starting material for phytoplasma detection. Leaf tissues (10 g) of 101 samples were pulverized in liquid nitrogen, and for total DNA extraction we started from 0.5 g following the CTAB method [29]. The total DNA, diluted to 1:10, was then used for the molecular detection of ChMV and phytoplasma.

For the detection of ChMV, the DNA was amplified in a 50 µL volume containing 10 mM Tris-HCl (pH 8.5), 2 mM MgCl$_2$, 50 mM KCl, 0.2 mM dNTPs, forward and reverse primers (BadCh 5860F/6093R) [19] at 1 µM each, and 1 U of GoTaq (Promega, Madison,

WI, USA). After an initial denaturation step at 95 °C for 4 min, 35 cycles were set at 94 °C for 30 s, 60 °C for 30 s, and 72 °C for 30 s, followed by a final extension step of 10 min at 72 °C [19].

For the detection of phytoplasma infection, a nested PCR assay with two phytoplasma universal primer sets was performed. P1/P7 primers [30,31] were used in the first PCR, and after a 1:30 dilution of the first PCR product, R16F2n/R16R2 [32,33] were used in the second run. The DNA was amplified in a 20 μL volume containing 10 mM Tris-HCl (pH 8.5), 2 mM MgCl$_2$, 50 mM KCl, 0.2 mM dNTPs, forward and reverse primers at 0.5 μM each, 1 U of GoTaq (Promega), and 2 μL of template. The PCR conditions consisted of an initial denaturation step of 95 °C for 2 min, followed by 40 cycles of 95 °C for 30 s, 53 °C (P1/P7 primers) or 50 °C (R16F2n/R16R2 primers) for 30 s, and 72 °C for 1 min and a final extension step of 72 °C for 5 min [34].

The amplified products were analyzed by electrophoresis on 1.5% agarose gels stained with Midori Green Advance (Nippon Genetics, Düren, Germany) with a 100 bp DNA ladder (New England Biolabs, Ipswich, MA, USA) used as the marker. After electrophoresis in 0.5× TAE buffer, a picture was captured by the Gel Doc™ XR imaging system (Bio-Rad, Hercules, CA, USA).

### 2.3.2. Quantification of Nutrients in Leaf Samples and Chemical Analysis of Soils

For the principal macro- and micronutrients (P, K, Ca, S, Fe, Mn, B, Cu, and Zn), twenty leaves for each sampled tree were prepared, acid-digested, and analyzed with an optical emission spectrometer, as reported in Bertoldi et al. [18]. The total N was analyzed using an elemental analyzer (Primacs SNC100, Skalar Analytical B.V., Breda, The Netherlands) with a combustion temperature of 1200 °C. The analysis of the soil reaction, active lime, and available fractions of Mn and Fe were performed on air-dried and sieved (<2 mm) soil samples, as described in Bertoldi et al. [18]. The total organic and inorganic carbon (total lime) were quantified with an elemental analyzer using a temperature ramping program. The two parameters were measured after combustion at 600 °C and 900 °C.

### 2.4. Statistical Analysis

According to the symptomatic frames (mosaic, yellowing, and symptomless) and the positivity to ChMV by PCR, we distinguished five different groups: (1) asymp-neg = no symptoms and negative virus detection; (2) asymp-pos = no symptoms but positive for virus detection; (3) mosaic-pos = symptoms of mosaic and positive for virus detection; (4) yellowing-neg = symptoms of nutrient deficiency and negative for virus detection; and (5) yellowing-pos = symptoms of nutrient deficiency and positive for virus detection. The two samples "mosaic symptomatic but virus negative" were not considered because there were too few to be representative of the variability.

Statistically significant differences between the five groups for the mineral nutrients in leaves, chemical parameters in soils, and SPAD measurements were highlighted by the Kruskal–Wallis test ($p < 0.05$). All statistical analyses were performed with Dell Statistica 64® version 13 (Dell Inc., Tulsa, OK, USA).

In addition, a principal component analysis (PCA) was computed to search for similarities among samples on the basis of their chemical-physical properties and highlight the correlations among the analyzed parameters. The data were pretreated with autoscaling, and samples with missing values were removed before the PCA computation, returning a dataset of 76 observations × 11 variables. MATLAB software (ver. MATLAB R2019b, The MathWorks, Massachusetts USA) was used for the PCA computation.

## 3. Results

### 3.1. Field Investigations

#### 3.1.1. Tree Visual Assessment in the Plots

A total of 19 sites were reported to be affected by symptomatic crowns, while two more were randomly selected as a control without symptoms during the surveys. Two nurseries

were visited because they reported yellowing on chestnut seedlings. Symptoms were recorded either in traditional chestnut orchards (12) or in new plantations (2). Symptomatic trees were also observed in both orchards and in the nearest mixed wood (4); another site was directly in a mixed wood (1) (Table 1). From the interviews with people who signaled the problems, most of these symptoms were of recent appearance, and only in a few cases, mainly nutrient deficiencies, could be estimated to have occurred for more than ten or fifteen years. In all cases, except one, the appearance of the virus was reported recently and occurred in the last five years.

On trees showing color alteration in the canopy and leaf shape anomalies, we were able to discriminate between two different symptomatologic frameworks: mosaic and yellowing symptoms (Figure 1).

The mosaic symptoms can be described as light and dark green patches on the leaves, accompanied by shoots with asymmetric blade deformation. This symptom is generally progressive. First, it is mild; then, along the season, the severity increases, and it is characterized by a bright mosaic, leaf curling, and the desiccation of the border of the blame (Figure 1A,B,E). The symptoms can involve a few shoots or most of the canopy.

The yellowing symptoms (described by Bertoldi et al. [18]) are characterized by the appearance of new leaves that are uniformly chlorotic with well-defined green bands along the veins. During the season, the leaves become almost withered and begin to be distorted (Figure 1C,D). The final stage shows leaves with a necrotic margin that are crinkled or curled. Sometimes the affected leaves also appear to be reduced in size. The symptoms can affect the whole crown or single branches. Interestingly, these symptoms appeared suddenly but then persisted on the trees in the following vegetative seasons. Affected trees have reduced production, but no mortality has been observed thus far.

During the surveys, we assessed the symptoms in 101 chestnut trees: 23 plants showed mosaic and deformation of the leaf blade, 38 showed yellowing symptoms, and 40 plants were selected because they were symptomless. The plants showing mosaic symptoms were recorded in Marche (10/18), Emilia Romagna (6/27), Toscana (4/13), and Trentino (3/36) (Table S1). Yellowing symptoms were recorded mostly in Toscana (5/13), Trentino (20/36), Alto Adige (5/7), and Emilia Romagna (8/27). The visual assessment conducted in the nurseries allowed us to record yellowing at site TN8 and mosaics at site AD1 (Table S1). Mosaic symptoms were recorded on old grafted trees, in some cases truly centenary and monumental; on younger grafted trees; on new plantations with grafted trees of 4–5 years; on coppice sprouts; and on natural regeneration in woods near the reported affected orchards. In the nursery, 1-year-old ungrafted seedlings showed symptoms. The same pattern of distribution of the damage was also recorded for the yellowing observed in both orchards and woods on subjects of different ages and different origins (old and young grafted trees as well as sprouts or ungrafted trees).

Mosaic was generally observed in a single or a few symptomatic plants, sometimes with few leaves of interest. Only at site M2 (Acquasanta Terme, Marche) did the disease affect a large area (approximately 40 hectares), involving both grafted trees and coppices. Chestnut blight was widespread in all surveyed areas (Table 2), but a clear predominance of healing and healed cankers was recorded. Virulent infections were limited to single branches or sprouts. Moreover, trees with symptoms of yellowing or viral presence showed no alteration in the general pattern of the prevalence of hypovirulence: no increase in virulent infection was observed, even on the most affected crowns, and hypovirulent infections were also present in these subjects. Generally, estimations of different canker rates showed hypovirulence rates ranging from 88 to 92% of the observed infections in the areas.

Ink disease mainly affected site M2, where in recent years it caused high levels of damage, with the deaths of old and young plants, both grafted trees and coppice sprouts. Suffering plants not showing yellowing or viral infection were described as ink-symptomatic in ER1 and ER3, while in the other sites the disease was not recorded during the surveys.

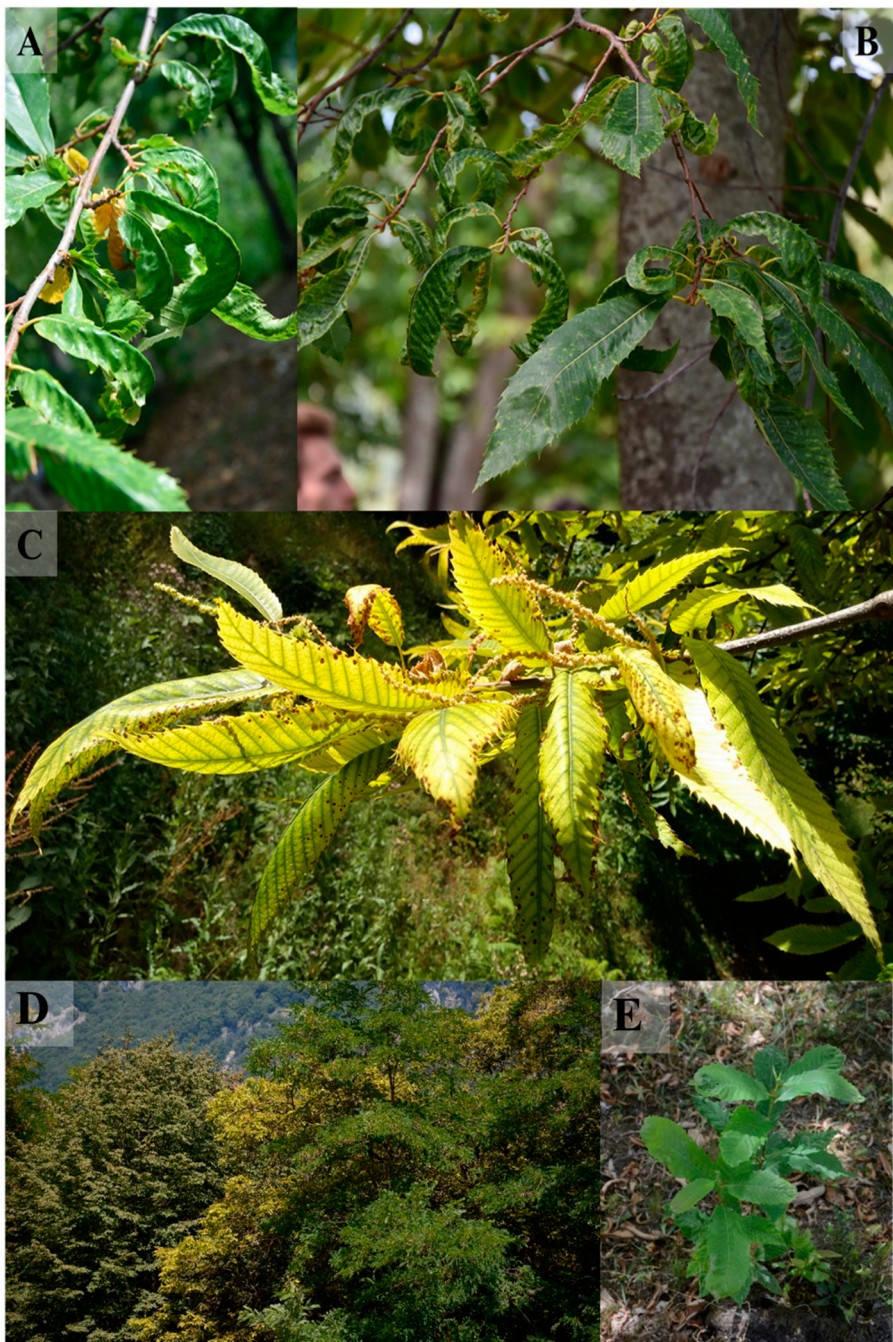

**Figure 1.** Mosaic (**A**,**B**) and yellowing symptoms (**C**) recorded in the Trentino Alto Adige, Toscana, Emilia Romagna, and Marche regions. Yellowing symptoms on chestnut in mixed wood (**D**). Natural regeneration affected by mosaic symptoms (**E**).

Gall wasps were recorded at all the sites but with a very low intensity: the number of galls generally ranged from sporadic (1 to 10 galls) to present (11–100), but more than 100 galls were never observed. The yellowing or virus-symptomatic trees did not show an increase in wasp attacks towards the healthy trees of the same area. Notably, in the few examined galls, the presence of parasitoid larvae was clearly visible in all areas.

No appearance of significant attacks on the other foliar disease was detected at any of the sites during the surveys.

**Table 2.** Main phytosanitary issues recorded in the chestnut stands.

| Site ID | Location | Main Phytosanitary Problems | | | | | |
|---|---|---|---|---|---|---|---|
| | | Mosaic | Yellowing | Blight | Ink Disease | ACGW | Other |
| M1 | Venamartello | 1 | 1 | 4 | 0 | 2 | 0 |
| M2 | Acquasanta Terme–Umito | 4 | 0 | 4 | 4 | 2 | 0 |
| ER1 | Selva la Maddalena | 0 | 2 | 4 | 1 | 2 | 0 |
| ER2 | Loiano | 1 | 1 | 4 | 0 | 2 | 0 |
| ER3 | Ca' di Balloni | 1 | 0 | 4 | 1 | 2 | 0 |
| ER4 | Gragnano | 2 | 0 | 4 | 0 | 2 | 0 |
| ER5 | Malarbergo 1 | 1 | 0 | 4 | 0 | 2 | 0 |
| ER6 | Malalbergo 2 | 1 | 0 | 4 | 0 | 2 | 0 |
| ER7 | Alberelli | 1 | 0 | 4 | 0 | 2 | 0 |
| ER8 | Monteombraro Lavezze | 1 | 1 | 4 | 0 | 2 | 0 |
| ER9 | Monteombraro Caseificio | 1 | 0 | 4 | 0 | 2 | 0 |
| ER10 | Montese | 0 | 1 | 4 | 0 | 2 | 0 |
| TO1 | Castagneto | 2 | 2 | 4 | 0 | 2 | 0 |
| TO2 | Gaggiolo | 1 | 0 | 4 | 0 | 2 | 0 |
| TN1 | Mezzolombarbdo | 0 | 0 | 4 | 0 | 2 | 0 |
| TN2 | Drena | 0 | 1 | 4 | 0 | 2 | 0 |
| TN3 | Pranzo 1 | 1 | 1 | 4 | 0 | 2 | 0 |
| TN4 | Pranzo 2 | 0 | 1 | 4 | 0 | 2 | 0 |
| TN5 | Pranzo 3 | 0 | 1 | 4 | 0 | 2 | 0 |
| TN6 | Campi | 0 | 1 | 4 | 0 | 2 | 0 |
| TN7 | Valternigo | 2 | 2 | 0 | 0 | 2 | 0 |
| TN8 | Nursery San Michele | 1 | 0 | 0 | 0 | 0 | 0 |
| AD1 | Nursery Piccolongo | 0 | 1 | 0 | 0 | 0 | 0 |

0 = not present, 1 = sporadic presence (1–5 trees affected), 2 = medium presence (6–10 trees affected), 3 = high (11–15 trees), 4 = very high (over 16 trees) (ACGW= Asian chestnut gall wasp).

### 3.1.2. SPAD Measurements

The surveyed chestnut trees showed a wide range of chlorophyll content, with a minimum SPAD value of 12.6 and a maximum of 41.6. Nevertheless, the SPAD values observed within each group were quite homogeneous, with medians of 23.2 (min–max 11.5–30.6), 38.2 (32.2–41.2), and 32.0 (21.7–39.7) in symptomatic, asymptomatic, and recovered trees, respectively (Figure 2). Highly significant differences were noted by the Kruskal–Wallis test ($p < 0.001$) for symptomatic vs. asymptomatic leaves. The data suggest that an SPAD value of 30 represents a threshold separating symptomatic (mosaic and yellowing) and asymptomatic trees. Recovered trees are in an intermediate position, with SPAD values partially overlapping those of the other groups.

### 3.2. Laboratory Tests

#### 3.2.1. Molecular Detection of Phytoplasma and Chestnut Mosaic Virus in Chestnut Samples

The samples collected according to the different symptomatologies were analyzed by molecular tools to detect the associated causal agents. All 101 samples, corresponding to trees with mosaic and/or yellowing symptoms and asymptomatic ones, tested negative for phytoplasma infection.

We detected ChMV in 21/23 (91.3%) samples with mosaic symptoms, in 12/38 (31.6%) with nutrient deficiency, and in 12/40 (30.0%) with no symptoms (Table 3). Samples that were positive during molecular detection showed a specific amplicon of approximately 230 bp (Figure 3).

In twelve situations, the samples showing yellowing due to nutrient deficiency were positive for the virus. In particular, viral symptoms can be masked by the symptoms caused by nutrient deficiency, especially in an early infection stage; for this reason, molecular tools could contribute to clearly discriminating ambiguous situations.

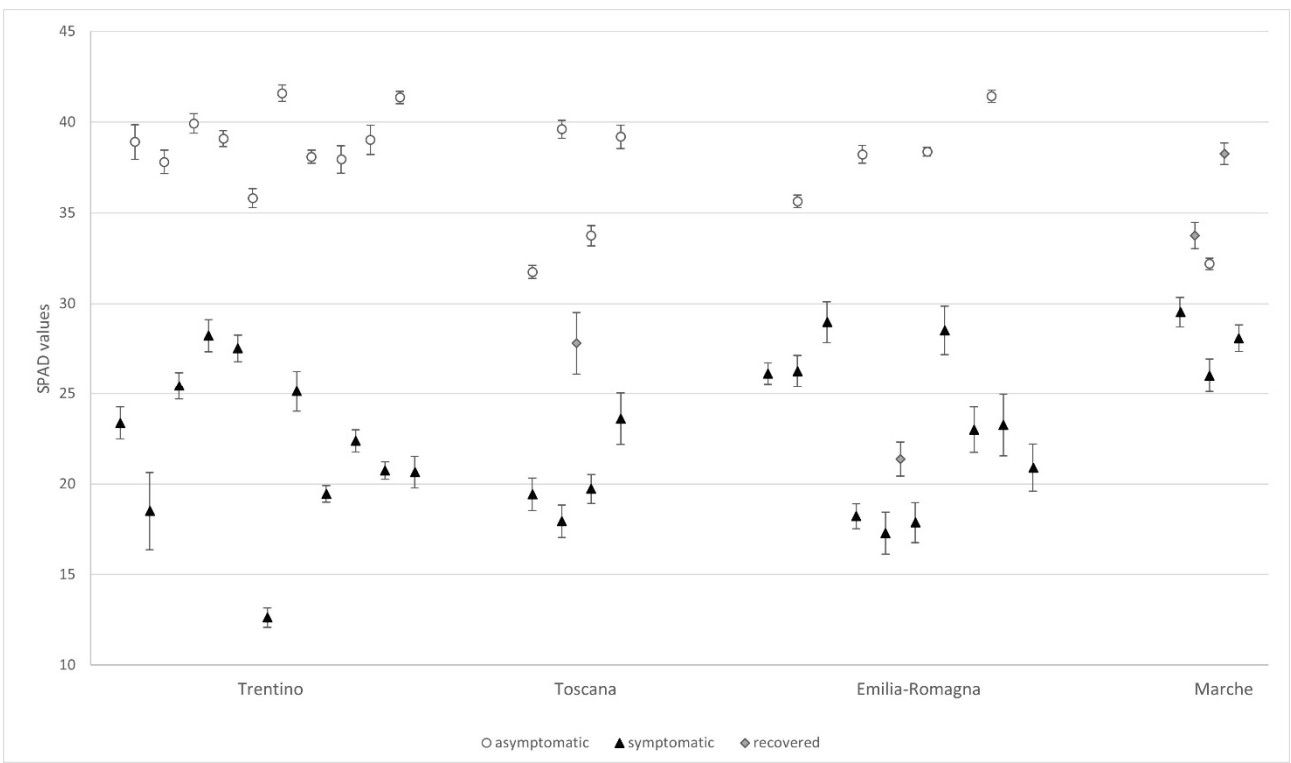

**Figure 2.** Chlorophyll contents (mean values ± standard errors) estimated by an SPAD-502 in the symptomatic and asymptomatic leaves of the surveyed chestnuts. Four recovered trees were also examined.

**Table 3.** Molecular detection of ChMV carried out on mosaic, yellowing, and asymptomatic samples.

| Symptoms | No. Samples | Molecular Analysis | |
|---|---|---|---|
| | | ChMV (%) | Phytoplasma (%) |
| Mosaic | 23 | 20 (91.3) | 0 |
| Nutrient deficiency | 38 | 12 (31.6) | 0 |
| No symptom | 40 | 12 (30.00) | 0 |
| Total | 101 | 45 | 0 |

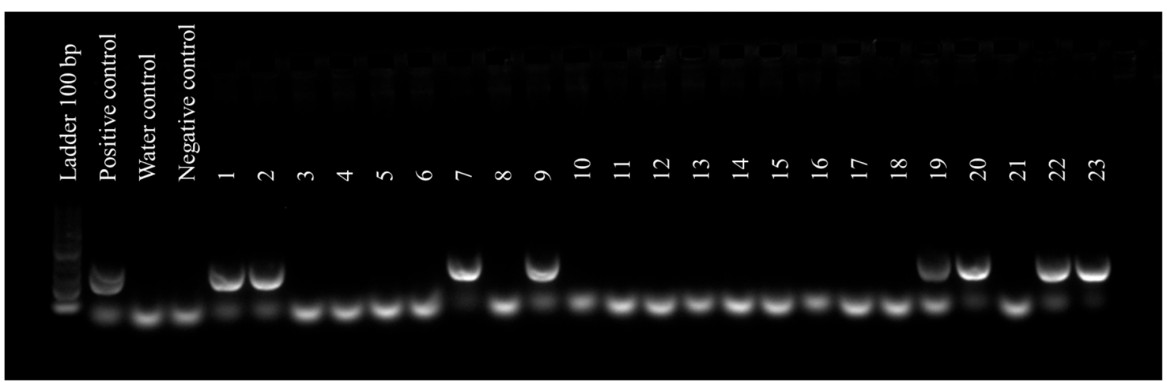

**Figure 3.** Molecular detection of ChMV was conducted on samples showing yellowing, mosaic, and no symptoms. Rows 1–2: asymptomatic/+; 3–6: healthy; 7 and 9: nutrient deficiency/+; 8 and 10–17 nutrient deficiency/−; 18 and 21: mosaic/−; 19–20 and 22–23: mosaic/+.

3.2.2. Nutrient Elements in Leaf Samples and Chemical Analysis of Soils

Regarding leaf mineral contents, the observed ranges are presented in Table S2. The Mn level was lower in samples presenting nutrient deficiency symptoms (4 and 5 in Figure 4)

than in the others, as previously observed [18]. Yellowing leaves (groups 4 and 5) also showed a tendency for lower contents of Fe, even if the difference between nutrient-deficient leaves and healthy leaves was not statistically significant for nutrient-deficient/virus-negative leaves (4) only. In contrast, the presence of the virus did not seem to influence the Mn level in leaves. The mineral leaf contents of samples infected by ChMV were never significantly different from those of healthy leaves. The lower content tendency of Ca and Mg could be related to the soil reaction of the growth soil for these trees, which were mainly distributed in acidic soils in Marche and Emilia Romagna and were not directly linked to viral infection.

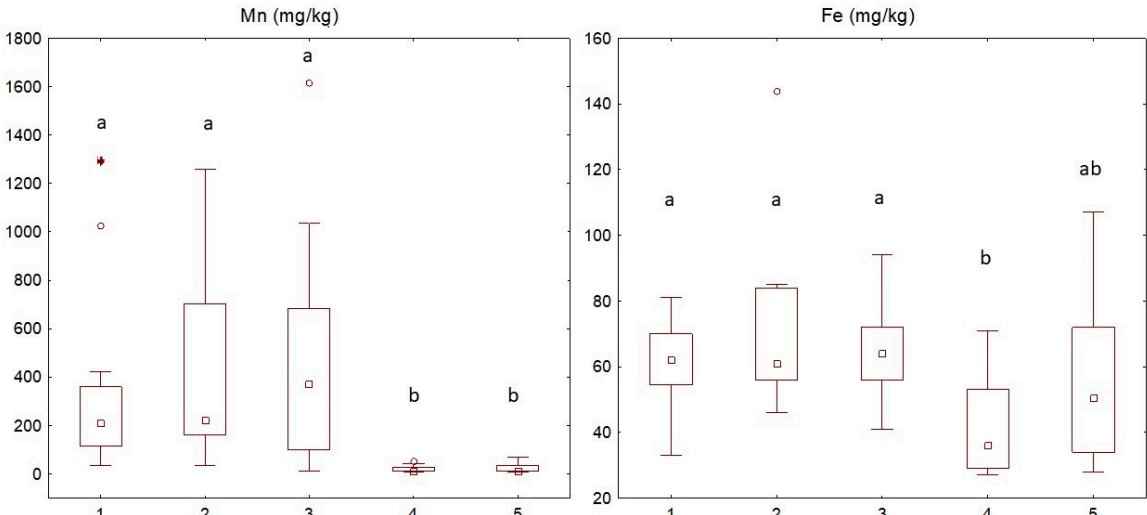

**Figure 4.** Box plot of the distribution of the Mn and Fe contents in leaves in the different groups. Groups: 1—Healthy = no symptoms and virus-negative; 2—Asymptomatic/+ = no symptoms but virus-positive; 3—Mosaic/+ = symptoms of mosaic and virus-positive; 4—Nutrient deficiency/− = symptoms of nutrient deficiency and virus-negative; 5—Nutrient deficiency/+ = symptoms of nutrient deficiency and virus-positive. Asterisks: extremes; circles: outliers; central square: median value. Different letters indicate significant differences ($p < 0.05$).

No significant differences were observed between the five categories for the chemical parameters of the soil by means of the Kruskal–Wallis test. The available Mn in the soil of nutrient-deficient chestnuts was not lower than that of healthy plants, despite the lower Mn content in the leaves. In particular, for the growth soil of nutrient deficiency/+ and nutrient deficiency/− plants, the available Mn ranged between 3.8 and 18.3 mg/kg, whereas for healthy plants it ranged from 5.4 to 32.9 mg/kg (Table S3). As reported in Bertoldi et al., growth in calcareous and alkaline soils seems to favor nutrient deficiency [18]. At the Loiano (ER1) and Selva la Maddalena (ER2) sites, the soil of the growth of yellowing trees showed an alkaline or subalkaline reaction, whereas the soil of near healthy trees showed an acidic or neutral reaction, indicating a strict relationship between suffering trees and soil or substrate conditions. For one of the nurseries with Mn deficiencies, irrigation water showed a clear alkaline composition with a high pH (data not shown).

3.2.3. Trends of Mineral Element Contents in Leaves during the Season

The results highlight that the average contents of Mn, Mg, B, K, and P were very distinct for the separate sites, probably because for these elements the leaf content derives from the soil content and is bound to the geomorphological characteristics of the site (Table S4). In detail, considering all samples for each site, K and Mn were significantly higher at site 1 with respect to sites 2 and 3; for Mg, site 3 was higher than at sites 1 and 2. For P, site 1 was higher than site 2, whereas site 3 was higher than site 2 for B. The difference between the samples of the three sites was higher than 40%, and for Mn it was higher than

100%. The wide range of Mn content for healthy leaves was already reported in Bertoldi et al. [18].

The leaf contents of Ca, Mn, P, B, and Fe increased from June to October, and the Cu content decreased, while for the other studied elements, the contents remained almost stable during the vegetative season. In detail, Ca showed on average an increase of 82% (min–max: 65%–100%), Mn showed on average an increase of 59% (33%–90%), P showed on average an increase of 37% (21%–56%), B showed on average an increase of 29% (8%–47%), and Fe showed on average an increase of 25% (9%–51%), whereas Cu decreased, on average, by 29% (25%–33%).

In some cases, yellowing symptoms were more evident at the beginning of the season and then faded. This could be partially explained by the physiological increase in Mn levels in leaves.

### 3.2.4. PCA Elaboration

A PCA was carried out to investigate the dataset variability and to look for the sample distribution/grouping based on pattern similarity. An outlier sample, a nursery sample with very different data from the others, probably due to fertilization, was removed before the computation. The final dataset consisted of 79 samples (group 1: 20 samples; group 2: 11 samples; group 3: 19 samples; group 4: 19 samples; and group 5: 10 samples). The left side of Figure 5 reports the PCA biplot scores (dots) and the loadings (lines), showing how strongly each chemical parameter (original variable) influences the principal components (PCs) (new variables). Since PCs are uncorrelated, the PCA loading plot is able to highlight the correlations among the original variables. Mn, Fe, and B were negatively correlated with Ca and Mg. The former are responsible for the location of the samples of groups 2 and 3 in the upper left part of the score plot, while Ca and Mg tend to move the samples of groups 4 and 5 to the bottom part of the score plot. The samples are colored in accordance with the type of infection.

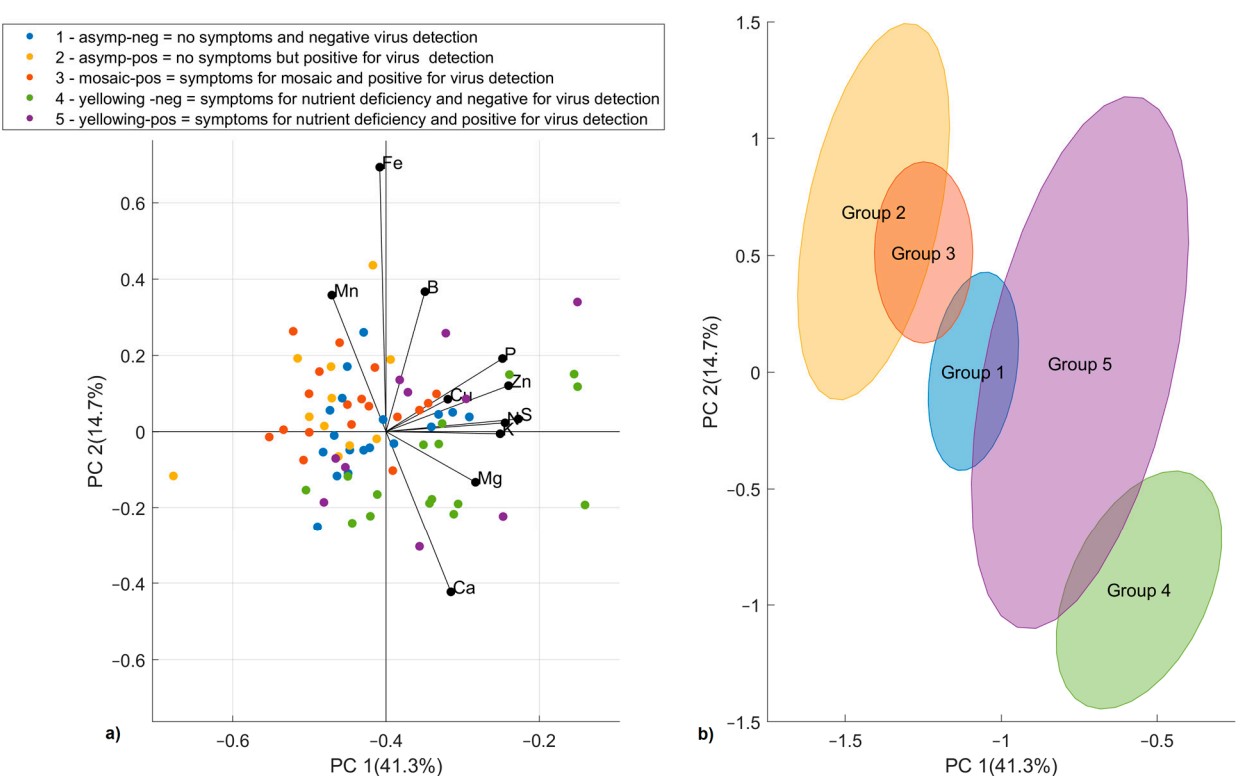

**Figure 5.** PCA biplot of first two principal components, with samples colored according to the type of infection (**a**). PCA score plot with standard error ellipses for each type of infection (**b**).

To obtain a clearer picture, confidence ellipses were drawn around each group of samples using the local PCA on the scores (Figure 5, right side). As can be noted, samples with nutrition deficiency have negative values of PC2. Instead, groups 2 and 3 have positive values of PC2 and negative values of PC1. PC1 tends to divide samples as virus-negative or virus-positive.

## 4. Discussion

This research, triggered by the reports of chestnut growers across northern-central Italy, confirmed the presence of canopy disorders at all reported sites and allowed us to observe the problem in nurseries and on coppices or mixed wood, with symptomatic trees occasionally determined during the survey. The two kinds of symptoms described in this study could have a larger distribution and could also be present in other chestnut woods, allowing them to remain unnoticed. In the past, anomalies defined as "chestnut yellows" were reported in the literature and were putatively associated with phytoplasma [15–17,35–39]. On the other hand, the symptomatology of "chestnut mosaic" was associated with a virus [20,21,37,40,41].

According to our data, which confirm previous studies [18,19], the presence of phytoplasma can be ruled out because all analyzed samples tested negative. In contrast, chestnut yellowing was associated with Mn deficiency. Regarding the leaf mineral contents, the Mn level was lower in leaves showing nutrient deficiency symptoms, even if the available Mn showed no significantly different contents in soils collected below yellowing or healthy trees. For Mn deficiencies, the local site condition and probably the alkaline contents of circulating water seem to be the main factors causing the appearance of yellowing. As a result, these kinds of symptoms appear to be strictly related to the site and are not able to spread on large surfaces. Most foci of the disorders were very localized and were limited to a few plants, generally groups of up to 10 subjects or stripes of wood. As reported by Bertoldi et al. [18] in the Trentino areas, yellowing affected large stripes of orchards and wood or coppices in 2014, but in the survey conducted in this work, most old foci were reduced to a few sporadic symptomatic trees and was highly localized, therefore suggesting the ability to recover for most of the affected trees. The sudden appearance of yellowing, even in old trees, could be related to some stress, such as the long-term effects of the ACGW on organic soil or the impact of climate change on precipitation patterns, as hypothesized by Bertoldi et al. [18].

On the other hand, the chestnut mosaic was generally observed in a single or a few symptomatic plants, sometimes with few leaves of interest. Only in M2, Acquasanta Terme (Marche), was the disease reported on a large surface (approximately 40 hectares), affecting both grafted trees and coppices. At this site, the involved area has progressively enlarged since the first symptom records [41]. According to the molecular detection conducted on the 101 samples collected in different Italian regions, ChMV infected chestnut stands in the Marche region, where the full-length genome of this Badnavirus was recently sequenced [19], but it was recorded not only according to symptoms but also by molecular tools in the Apennines and pre-Alps. The mineral leaf contents of samples infected by ChMV were never significantly different from those in healthy leaves. This demonstrates that mosaic symptoms are induced by the virus and are not related to mineral deficiencies in the leaf or the soil. ChMV is mainly transmitted by grafting [14], but experimentally and likely naturally it can also be transmited through the aphid *Myzocallis castanicola* Baker [21,42–45]. The germplasm may also play an important role, as demonstrated in previous studies where latent ChMV infections were detected by biological indexing in many symptomless *C. sativa* varieties or *C. sativa x Castanea crenata* Siebold and Zucc. hybrids [46].

The appearance of the virus on nursery seedlings also needs to be further investigated to avoid the possibility of propagation with this material.

In our study, both viral infection and Mn deficiency negatively affected the efficiency of photosynthesis, as clearly demonstrated by the SPAD analysis, in agreement with other

studies [18,47]. The possibility of some actual stress on the affected trees is real; even if no dead trees or regeneration seedlings have been observed thus far, a prolonged suffering of the crowns could be dangerous. To date, both chestnut blight and ACGW seem not to be influenced by yellowing or mosaic disease, but local situations such as those reported for M2 suggest a possible link between viral infection and a large-scale spread of ink disease. Moreover, in the same site where the presence of the damage has been recorded since the 1990s, growers reported heavy effects on fruit production. Both these points need to be further investigated. Because of the widespread presence of *Phytophthorae* species in chestnut stands [48], it is conceivable that reduced photosynthesis due to viral presence can decrease the resistance of roots to pathogen attacks. The observed prevalence of hypovirulence in chestnut blight and the low level of ACGW attacks in all surveyed sites suggest that the biological control of these problematics is not yet affected by the virus or deficiency.

The recent recording of mosaic symptoms in most of the surveyed sites, symptoms on ancient trees, and the detection of latent infections on asymptomatic trees, as noted at sites ER5, ER6, and TO2, raises the question of whether the virus has recently arrived or was latent for a long period and finally induced symptoms. In addition, the vigor of trees and optimal climatic and pedoclimatic conditions can contribute to extending the latency period of the virus. In the context of climate change, both viral spread and virulence can be increased, with the risk of severe outbreaks [49]. Chestnut, both in forests and in traditional orchards, can be considered the archetype of general resilience, but cumulative stress conditions can induce generalized tree suffering that could have broken the latency of ChMV. In sporadic situations, the samples showing nutrient deficiency were positive for the virus. In particular, as shown by the PCA, viral symptoms can be masked by the symptoms caused by nutrient deficiency, especially in an early infection stage. For this reason, molecular tools and chemical analyses of leaves could contribute to discriminating ambiguous situations.

## 5. Conclusions

We can conclude that both problems described in this study are present in Italian chestnut woods, with very local effects. A possible relationship with other stress factors could be supposed, and their impact and spread need to be further monitored and investigated in the coming years. A control strategy can be proposed only in chestnut orchards and only for Mn deficiency through fertilization, while for viral infection only general good practices in grafting and the sanitary certification of seedlings could be applied to avoid undesired spread. In the more natural context of coppices and chestnut woods, monitoring could be achieved in relation to an evaluation of the ecophysiological effects on trees; on this point more data are desirable. At present, the described symptoms seem to be a limited problem for orchards and chestnut woods, but the increase in new reports suggests that these symptoms need to be considered an emerging issue whose impact is still to be studied and understood.

**Supplementary Materials:** The following supporting information can be downloaded at: https://www.mdpi.com/article/10.3390/f13111894/s1, Table S1: Samples collected during the surveys in chestnut stands located in Marche, Emilia Romagna, Toscana, and Trentino-Alto Adige and analyzed by molecular tools for phytoplasma and Chestnut mosaic virus; Table S2: Range (minimum–median–maximum) of mineral nutrient contents for leaves of the five recognized categories; Table S3: Range (minimum–maximum) of chemical parameters in the growth soil of the five recognized categories of plants; Table S4: Mineral contents in different months for the healthy leaves collected at three sites.

**Author Contributions:** S.M.: conceptualization, methodology, supervision, writing—review and editing, and founding. D.B.: writing—original draft, methodology, and chemical analysis. F.P.: writing—original draft, methodology, and molecular detection. M.M.: statistical analysis. G.R.: writing—review and editing, and founding. G.M.: resources, conceptualization, methodology, writing—review and editing, and supervision. All authors have read and agreed to the published version of the manuscript.

**Funding:** This work was partially funded by the UNIVPM project "Emergent disease for plant forests".

**Institutional Review Board Statement:** Not applicable.

**Informed Consent Statement:** Not applicable.

**Data Availability Statement:** Not applicable.

**Acknowledgments:** We would like to thank Ascenzio Santini, Renzo Panzacchi, Stefano Fogacci, Nerio Pifferi, and Emanuele Piani for supporting this research during the selection of the study areas.

**Conflicts of Interest:** The authors declare that they have no known competing financial interests or personal relationships that could have appeared to influence the work reported in this paper.

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
