# Peer review of "New Symptoms in Castanea sativa Stands in Italy: Chestnut Mosaic Virus and Nutrient Deficiency"

_forests, doi:10.3390/f13111894_

Round 1

Reviewer 1 Report

Dear Editor, the manuscript regards new symptoms of chestnuts in Italian orchards and stand. The manuscript is clear in all sections. However, I some changes should be considered

1)     In the introductions some important references are missed as (1) Vettraino, A.M., Morel, O., Perlerou, C. et al. Occurrence and distribution of Phytophthora species in European chestnut stands, and their association with Ink Disease and crown decline. Eur J Plant Pathol 111, 169–180 (2005). https://doi.org/10.1007/s10658-004-1882-0 2) AM Vettraino G Natili N Anselmi A Vannini (2001) Recovery and pathogenicity of Phytophthora species associated with a resurgence of ink disease on Castanea sativa in Italy Plant Pathology 50 90–96 10.1046/j.1365-3059.2001.00528.x).

2)      authors did not considered previous study on similar chestnut symptoms done by Vettraino et al (Vettraino et al. 2005-A NEW TRANSMISSIBLE SYMPTOMOLOGY ON SWEET CHESTNUT IN ITALYActaHortic; 1)     Sweet &Barbara 1978 A yellow mosaic disease of horse chestnut (Aesculus spp.) caused by apple mosaic virus- Annals of Applied Biology

3)     The statistical section needs to be integrated with the explanation of all the analysis done and reported in the text.

Moreover:

Line 81: how the interview has been done, which information have been asked ?

Line 89: provide more information about the number of selected orchards or coppice and report the results considering the management strategies of the sites survied

Line 89 Report in the text that you recorded data considered in Table 1

Line 100: July 2020?

Line 101 change in symptoms associated to chestnut mosaic disease

Line 101 change “the spread “ in the presence

Line 103: how did you chose the area to define the circle?

Line 104-117 : how authors considered the presence of different diseases, globally? For each group of symptoms  different approach has been used, a visual scale, presence/absence , intensity of the attacks. If all those symptoms have ben considered comments should be done in the discussion section

Line 117 please indicate the name of the authors (3)

Line 148…add: as described below

Line 157-158 delete Before starting any laboratory analysis, a picture was taken for each sample to record

the symptoms, because already wrote

Lline 170-178 move the sentences after line 164

Line 229 delete the citation (19)

Line 316: what authors mean with the term “sporadic”?

Line 339: specify which statistic analysis has been done

Line 351; in the section M&M it is not clear that the survey has been done for 1 year, and how many times/year?

Line 385 please indicate the PCA stress values

Author Response

sorry but my computer doesn't support the paste of the  answers.

so we have to add a file 

Author Response

sorry but my computer doesn't support the paste of the answers

so we upload the corrispondent file

Round 2

Reviewer 2 Report

All the issues mentioned previously have been modified